# State of the Art in the Diagnosis and Assessment of Oral Malignant and Potentially Malignant Disorders: Present Insights and Future Outlook—An Overview

**DOI:** 10.3390/bioengineering11030228

**Published:** 2024-02-28

**Authors:** Dardo Menditti, Mario Santagata, David Guida, Roberta Magliulo, Giovanni Maria D’Antonio, Samuel Staglianò, Ciro Emiliano Boschetti

**Affiliations:** 1Multidisciplinary Department of Medical-Surgical and Dental Specialties, University of Campania “Luigi Vanvitelli”, 80138 Naples, Italy; dardo.menditti@unicampania.it (D.M.); mario.santagata@unicampania.it (M.S.); roberta.magliulo@studenti.unicampania.it (R.M.); samuel.stagliano@studenti.unicampania.it (S.S.); ciroemiliano.boschetti@unicampania.it (C.E.B.); 2Division of Plastic and Reconstructive Surgery, Department of Surgical, Oncological and Oral Sciences, University of Palermo, 90128 Palermo, Italy; giovannimariadantonio@gmail.com

**Keywords:** Oral Potentially Malignant Disorders, OPMD, Oral Squamous Cell Carcinoma, OSCC, oral cancer, imaging technique, biomarkers

## Abstract

Oral Potentially Malignant Disorder (OPMD) is a significant concern for clinicians due to the risk of malignant transformation. Oral Squamous Cell Carcinoma (OSCC) is a common type of cancer with a low survival rate, causing over 200,000 new cases globally each year. Despite advancements in diagnosis and treatment, the five-year survival rate for OSCC patients remains under 50%. Early diagnosis can greatly improve the chances of survival. Therefore, understanding the development and transformation of OSCC and developing new diagnostic methods is crucial. The field of oral medicine has been advanced by technological and molecular innovations, leading to the integration of new medical technologies into dental practice. This study aims to outline the potential role of non-invasive imaging techniques and molecular signatures for the early detection of Oral Malignant and Potentially Malignant Disorders.

## 1. Introduction

The oral cavity constitutes a complex biological system, both from a histological point of view and from a functional–biomechanical point of view [1]. The oral system is an organ able to perform certain functions such as chewing, swallowing, phonation, breathing, facial expressions; these functions are made possible mainly by a pair of joints: the tempo-mandibular joints and naturally by the related muscles [2]. These anatomical structures can be affected by various kinds of disorders, including genetic, congenital, inflammatory and traumatic events [3,4,5]. Moreover, the niche of the oral antrum represents a favorable domain for the invasion of external agents from the external environment [6]. The coexistence between external microbial agents and the host is guaranteed by a constant balance of the oral habitat, nowadays also known as oral microbiome [7]. An alteration of the microbial flora, due to various intrinsic and extrinsic factors, may determine the phenomenon of dysbiosis [8,9,10]. The mechanisms involved in oral dysbiosis have shown to lead to most of the dental and periodontal tissues of the oral cavity, which recognize microbiological agents—viral, bacterial and fungal—as the main nosological factors [11,12,13]. Oral mucosal soft tissues, instead, are mainly affected by mycotic superinfections, with a greater prevalence for frail patients, immunosuppressed subjects and prosthesis wearers [14,15,16]. Furthermore, a large group of viral agents is involved in the etiological mechanism of a wide spectrum of oral mucosa diseases, ranging from acute to chronic disorders [17,18,19,20,21].

Pathological cytoarchitectural changes in the tissue layers of the oral mucosa are seen in a wide range of disorders, ranging from benign [22] to potentially malignant and properly malignant conditions [23,24,25,26], up to the involvement of rare syndromic manifestations involving multiple areas of the body [27,28]. Benign disorders affecting the oral mucosa represent a large group of pathological manifestations, especially autoimmune, viral and neuropsychiatric diseases [29,30,31,32,33,34]. However, the oral cavity is linked to the possible occurrence of malignant lesions, and 90% of them are represented by Oral Squamous Cell Carcinoma (OSCC), the 16th most common worldwide cancer [35].

Furthermore, there are several pathological entities involving oral mucosa that may show an increased risk of malignant transformation, which are also known as Oral Potentially Malignant Disorders (OPMD), and they are represented by manifestations such as Oral Leukoplakia, Oral Eritroplakia and Proliferative Verrucous Leukoplakia [36,37].

The most demanding challenge for clinicians involves the early detection and treatment of the OPMD manifestations, in order to prevent malignancy transformation [38,39]. The strongest efforts, to date, are focused on all the possible strategies able to achieve a time-saving diagnosis, ranging from biomarkers to imaging techniques [40,41,42,43]. The purpose of this overview is to describe a broad picture of oral cancerous and precancerous entities, and to depict the current approaches and the future perspectives to achieve a quick diagnosis. The aim of this study was to examine the available knowledge in the literature on the potential use of technology and biomarkers or accurately identifying and categorizing malignant or potentially malignant oral lesions at an early stage.

## 2. Oral Potentially Malignant Disorders 

Oral Potentially Malignant Disorders (OPDMs) represent a wide scenario of pathological manifestations, which may not include the onset of oral cavity malignant neoplastic lesions; this nomenclature has been asserted because not all disorders described under this term may transform into cancer [38].

### 2.1. Oral Leukoplakia, Proliferative Verrucous Leukoplakia and Erythroplakia

One of the most frequently encountered lesions classified as an OPMD in the oral cavity is Oral Leukoplakia (OL) [44] (Figure 1). A smoking habit has shown to be recurrently associated with leukoplakia in more than 80% of cases; OL represents a non-scraping white lesion and it is defined by the World Health Organization (WHO) as a “white plaque of questionable risk having excluded (other) known diseases or disorders that carry no increased risk for cancer” [45]. From a clinical point of view, there are different varieties for this entity: homogeneous leukoplakia presents as a flat and uniform white plaque with well-defined margins (at least one). Non-homogeneous leukoplakia involves spots of erythema surrounded by areas of nodularity and verrucousity [46]. Oral Proliferative Verrucous Leukoplakia (PVL) is a distinguishing subset of non-homogenous leukoplakia more common in females and less associated with a smoking habit. PVLs may involve a single extensive area but are often multifocal and frequently occur on the gingiva, buccal mucosa, and tongue in both bordering and non-contiguous sites of the oral cavity [47,48]. The mechanisms underlying the transformation from hyperkeratosis or hyperplasia to various degrees of dysplasia and eventually carcinoma in situ and/or OSCC are still subject to ongoing debate. As a result, PVLs and leukoplakias necessitate multiple recurring biopsies at different sites. This ongoing need for biopsies is essential to continuously investigate the histotype and detect different grades of dysplasia or malignant transformation to OSCC [49,50,51].

Furthermore, the lesion called “Oral erythroplakia” (OE) has been defined by the WHO as “a fiery red patch that cannot be characterized clinically or pathologically as any other definable disease” and it has long been deemed as the oral mucosal lesion with the greatest potential for malignant transformation in the oral cavity [52]. OE does not occur on any traumatic, vascular, or inflammatory basis, but appears with comparable frequency in both sexes in middle-aged and elderly patients. A diagnostic biopsy is required to detect this potentially malignant lesion, more accurately analyze this histological subset and even prevent the risk of transformation [53].

### 2.2. Oral Lichen Planus

To date, experts adhere to the distinction between precancerous lesions and precancerous conditions for the main reason that the onset of a malignancy in the mouth of a patient known to have a precancerous lesion would correspond with the site of precancer, while in precancerous conditions, malignancy may originate in any anatomical site of the oral cavity [45]. 

One of the most recognized oral potentially malignant conditions is represented by Oral Lichen Planus (OLP), a chronic immunological inflammatory disease that usually affects middle-aged patients, particularly women [54]. This disorder may involve the skin and mucosal membranes; moreover, OLP is usually linked to various patterns and the subtypes can be categorized as follows: reticular form, which is the most common subtype, distinguished by Wickham striae and hyperkeratotic plaques or papules; atrophic–erosive form, with areas of ulceration usually associated with keratotic white striae, and also ulcerations [55]. The plaque form of Oral Lichen Planus can mimic Oral Leukoplakias, which highlights the importance of biopsy [56]. OLP generally involves the buccal mucosa, followed by the gingiva and tongue, usually with multiple, bilateral lesions. Erosive and atrophic OLP can provoke intense discomfort and inhibit speech and swallowing [55]. It is still debated whether Oral Lichen Planus can progress into Oral Squamous Cell Carcinoma: data concerning the transformation rates range between 0% and 12.5%, and no clinical or histopathological suggestion can support the prognosis of potential malignant transformation [57,58]. The erosive form bears the highest risk rate of malignant transformation, followed by atrophic OLP, while the reticular white type has the lowest risk rate [59,60]. 

### 2.3. Submucosal Fibrosis

Another characteristic oral potentially malignant condition is represented by Submucosal Fibrosis [61]. This condition presents as a chronic fibrotic lesion of the oral mucosa, possibly as the expression of a large wound overhealing, based on a chronic mechanical or chemical insult [62]. With a predominance in males, the most commonly affected sites are the buccal mucosa, the lingual margins, the lip, the palate, and the gums [63]. Clinically, Submucosal Fibrosis is characterized by a loss of elasticity in the affected tissues, with palpable fibrous bands that involve the mobility of the tongue and restrict mouth opening [64].

To date, Submucosal Fibrosis has been widely recognized to carry a malignant potential; an estimated transformation rate of 8% has been reported [65].

### 2.4. Actinic Keratosis

Actinic Keratosis (AK) is a common precancerous skin lesion that may also affect oral and peri-oral tissues, caused by long-term sun exposure. It appears as a rough, scaly patch on sun-exposed areas of the skin, such as the face, neck, arms, and hands. If left untreated, AK can develop into squamous cell cancer. Early detection and treatment, such as cryotherapy or topical medications, can prevent the progression of AK to malignant transformation. Regular use of sun protection, such as sunscreen and protective clothing, can also reduce the risk of developing AK. It is important to seek the advice of a healthcare professional for proper diagnosis and management of actinic keratosis [66].

### 2.5. Graft versus Host Disease

According to the WHO Classification of Tumours, 4th Edition (2017) [67], Oral Graft versus Host Disease (GVHD) is considered as a potentially malignant disorder, and it describes a condition that can occur after a bone marrow or stem cell transplant, when the transplanted cells attack the recipient’s body. In Oral GVHD, the mouth, including the lips, gums, tongue, and inside of the cheeks, is affected [68]. Symptoms may include redness, swelling, sores, and painful or difficulty in swallowing. Oral GVHD can be a severe and life-threatening condition, and early detection and prompt treatment are important for the best outcome. Treatment may include topical or systemic medications, and in severe cases, additional interventions such as surgery may be necessary. Regular monitoring by a healthcare professional is important for the management of Oral GVHD. Nevertheless, no Oral Potentially Malignant Disorder is considered as a compulsory forerunner to cancer; likewise, most Oral Potentially Malignant Disorders do not lead to cancer [69]. Hence, the most challenging struggle involves determining lesions that display a higher risk of transformation from those carrying a lower risk. Dysplastic features are supposed to be the most valuable indicator of potential malignant progression [70,71].

Nonetheless, a lesion with no sign of dysplasia carries the potential to transform into cancer, and there are no specific clinical signs directly associated with dysplastic features; therefore, a biopsy is invariably mandatory to assess whether dysplasia is present [72]. 

## 3. Oral Malignancies

Oral malignant neoplasms belong to the variety of head and neck cancers (H&N cancers), which represent a wide group of epithelial malignant tumors involving the lining mucosa of the nasal cavity and paranasal sinuses, nasopharynx, hypopharynx, larynx and trachea, oropharynx, oral cavity and salivary glands [73]. Although there is a large group of mostly rare malignant neoplasms affecting the oromaxilofacial area [74,75,76], Oral Squamous Cell Carcinoma (OSCC) represents a malignant neoplasm involving the oral cavity, leading to more than 90% of malignant tumors of the head and neck area, ranking as the 16th most common worldwide cancer [77], and it stands as the most common malignancy in South-East Asia (India, Sri Lanka, Pakistan, Bangladesh, and Taiwan) and the Pacific regions (Papua New Guinea and Melanesia) due to the betel chewing habit [78] (Figure 2). Tobacco and alcohol consumption represent two of the major risk factors for the development of oral cancer [79]; nevertheless, many other elements have been reported to increase the risk of OSCC, such as micronutrient depletion, hormonal, protein and enzyme imbalances [80,81,82,83,84], poor oral hygiene, chronic traumatism, and viruses [85,86]. Poor oral hygiene has been always associated with the risk of the development of periodontal disease, so current investigations are focusing on the possible association between the periodontal status of health and oral cancer risk [87]. The potential role of human papillomavirus (HPV) infection and its relationship with cancerogenesis are the most explored but still debated topics among all human viruses. Human papillomavirus infection has a potential role in carcinogenesis. To date, more than 100 genotypes of HPV are known, divided into low risk and high risk. The viral proteins E6 and E7 play a key role in pathogenesis, particularly by inhibiting the oncosuppressors p53 and Rb, with overexpression of the p16 protein. Data in the literature show that HPV16 represents about 86% of HPV+ oropharyngeal squamous cell carcinomas (OPSCC) cases, while the association between OSCC and human papillomavirus infection is still debated [88,89,90,91]. From a purely clinical point of view, various forms have been reported, including the endophytic form, infiltrative exophytic form, vegetative or papillary, ulcerative form and mixed form [92]. The recurrence and prognosis of OSCC have been deeply investigated in different studies and a close relationship with histological malignancy grading and different clinical parameters has been found [35,93,94]. Besides the recurrence of OSCC, survival and mortality rely on multiple and additional biological, histological, macroscopic, and microscopic aspects that have been analyzed in order to determine the causalities, support early diagnosis and develop suitable remedies for the different characteristics and grades of Oral Squamous Cell Carcinomas [95,96]. Site, size and thickness—or the depth of invasion—of the cancer have been the most examined prognostic factors for achieving the current survival rates. The closer the tumor origin is to the inner sites of the mouth, the lower the survival rates [97]; furthermore, the survival rate decreases in relation to the advanced involvement of regional lymph nodes [98]. To date, approximately 60% of OSCC lesions are detected in locally advanced stages, when the 5-year survival rate is <50–60% [99]. In addition, conventional histopathological data seem to be unsatisfactory in accurately foretelling the clinical evolution of OSCC [100]; consequently, the need for a label of molecular markers that can predictively demarcate which cancer will display an aggressive behavior and poorer prognosis should be highlighted [101,102,103]. Treatment options include surgery, radiation therapy, chemotherapy, and targeted therapy [93]. The type and extent of treatment will depend on the size and location of the tumor and the overall health of the patient. It is important to work with a healthcare team to develop an individualized treatment plan. Regular dental check-ups and oral cavity screenings can help detect OSCC in its early stages and increase the chances of successful treatment [67].

The early diagnosis of the asymptomatic first stages of oral cancer is still the pivotal and most demanding step to achieving an adequate clinical outcome in most patients.

## 4. Future Perspectives

Early diagnosis of oral malignant and potentially malignant lesions is crucial for improving the chances of successful treatment. Prompt detection of oral cancer can be aided through routine dental exams and screenings for the disease [78]. During a screening, a dentist or physician will examine the mouth, neck, face, and lymph nodes for signs of abnormalities. They may also feel for lumps or thick patches in the tissues and check for any changes in the way the teeth fit together when the mouth is closed [73]. In some cases, clinicians should be assisted by new technological instruments, such as a bright light or magnifying glass, in order to take a closer look at the tissues. If an abnormality is detected, the individual may be referred for further testing, such as a biopsy, to determine if it is cancerous [104]. Within this context, AJCC/UICC suggest performing HPV tests using techniques such as DNA/RNA HIS (in situ hybridization), DNA-PCR, p16 immunostaining or the more recent circulating tumor tissue-modified viral (TTMV)-HPV DNA plasma testing (still being studied). Indeed, the new TNM classification for head and neck cancers (8th edition) distinguishes between HPV-negative oropharyngeal cancers and HPV-positive oropharyngeal cancers (OPC) to improve patient communication, clinical management, risk stratification and disease monitoring [105,106]. Therefore, it is important for individuals to be aware of any changes in the oral tissues and seek medical attention if necessary.

The main works retrieved and included in this scoping review have been summarized and synthetized and are presented in Table 1.

The need to intercept premalignant lesions, and to identify them before they give rise to a malignant neoplasm, has directed efforts towards the search for increasingly effective and reliable innovative methodologies [104]. Following this pursuit, in the last few decades, the advancement of scientific technologies and molecular research have allowed us to explore markers and genetic and epigenetic factors and to define their link to early diagnosis and OSCC progression and prognosis, thus treating them as essential in future individual and tailor-made approaches [107,108,109]. In the future, advancements in technology and medical research may lead to improved methods for diagnosing oral cancer. For example, researchers are exploring the use of biomarkers, such as saliva and blood tests, to detect oral cancer at its earliest stages [110]. Artificial intelligence and machine learning may also play a role in oral cancer diagnosis by helping healthcare providers to identify and analyze patterns in large amounts of data [111]. Additionally, there is ongoing research into the use of devices such as handheld spectroscopy tools and portable imaging devices, which may make it easier for healthcare providers to perform oral cancer screenings in various settings. The future of oral cancer diagnosis is likely to involve a combination of these and other innovative technologies, with the goal of improving accuracy and early detection rates.

### 4.1. Biomarkers

Biomarkers for oral malignant and premalignant lesions refer to substances or characteristics in the body that indicate the presence of a disease or condition. These biomarkers represent a strategy target that may be involved in diagnosing and monitoring the evolution of these kinds of lesions in their early stages or to monitor the effectiveness of treatment [112]. Examples of oral cancer biomarkers include certain proteins, genes, or microRNA.

One of the major targets of interest has been the down-regulation of cell adhesion proteins, which has proven to contribute to OSCC progression [113]. Several authors have investigated E-cadherin and P-cadherin expression and reported their role as negative prognostic factors of OSCC, due to their aggressive biological behavior with a tendency to infiltrate and metastasize [70,80]. Likewise, several studies have been conducted focusing on the prognostic role of CD44 in oral carcinoma progression; according to Carinci et al., an under-expressed level of CD44 is related to a reduced survival rate and contrariwise, the over-expression of CD44 in primary tumors was compatible with a more extended survival outcome [114]. Following this path, the advancements of molecular analyses have led to the exploration of potential markers and genetic and epigenetic factors in an effort to elucidate their link to early diagnosis and OSCC progression and prognosis and use them as targets in future studies [83,85]. 

The identification of molecular markers that can accurately characterize lesions that will display aggressive behavior and worse prognosis has been enhanced with the improvement of imaging techniques, which have been introduced in oral medicine to evaluate their potential role in early diagnosing tumoral signs of alteration, thus lessening the need for biopsy [115,116,117,118].

It is important to note that biomarker research is still in its early stages and more research is needed to fully understand their role in oral cancer diagnosis and treatment.

Therefore, due to the lack of large-scale studies and insufficient data, the use of biomarkers alone is not enough to diagnose and follow up these lesions and biopsy remains mandatory [119].

### 4.2. Reflectance Confocal Microscopy (RCM)

The in vivo and real-time visualization of oral cavity tissue layers demonstrated its usefulness during the last few decades in the diagnosis of suspicious lesions, showing a considerable sensitivity both for OSCC and Oral Potentially Malignant Disorders [120,121]. A critical and deep investigation of the imaging technology’s applicability to potentially malignant disorders has been carried out. Reflectance Confocal Microscopy (RCM) represents one of the most interesting devices, based on a system composed by a diode laser at 830 nm and combined with a water immersion objective lens [122]. RCM offers optical scanning by horizontal planes, layer by layer, with the ability to deeply penetrate both ex vivo samples and in vivo tissues [123]. Interesting confocal images of dysplastic and neoplastic oral tissues were documented by Maitland et al., proving a strong correlation between RCM and histological sections [124]. Furthermore, Agozzino et al. exhibited remarkable data from their RCM experience with in vivo imaging of Oral Malignant Disorders; the authors reported confocal images of epithelial layers with atypia and polymorphic keratinocytes in Oral Squamous Cell Carcinoma, before its histological confirmation [125]. 

### 4.3. Optical Coherence Tomography (OCT) 

Similarly, based on interferometer technology, Optical Coherence Tomography (OCT) has proven to be a notable fit for oral cavity exploration [126]. OCT allows us to produce real-time cross-sectional sub-surface tissue images, conventionally explained as tomographic images, and generate real-time depth-resolved two-dimensional (2D) and three-dimensional (3D) images at superior spatial resolution (10–20 μm) and a satisfactory depth of penetration into the tissues [127]. In oral medicine, OCT has proven to provide several advantages over other imaging techniques, including its non-invasive nature, fast imaging speed, and high level of reproducibility [128]. These characteristics make OCT an important tool for ongoing monitoring of oral health and for guiding therapeutic interventions [118]. Additionally, OCT has the potential to aid in the early detection and diagnosis of oral diseases, potentially leading to improved outcomes for patients [129]. OCT technology has emphasized its useful diagnostic aid potential in oral disorders; a study by Wilder-Smith et al. showed 50 white or red potentially malignant lesions, explored by OCT, thus revealing a decisive correlation between OCT-based diagnostic pictures and the histological outcomes [130]. In addition, Heidari’s group highlighted the potential diagnostic accuracy of OCT scanning, through a diagnostic algorithm compared to clinical observations. The authors reported that OCT-based diagnosis was a possible resource to enhance the sensitivity and specificity of current procedures [131]. Furthermore, in a study by Sunny et al., they reported 125 OCT images of OSCCs. The OCT-based images highly correlated with the histological results, further revealing the ability to differentiate OSCC from non-dysplastic and dysplastic lesions and to differentiate oral epithelial dysplasia (OED) from non-dysplastic oral lesions [128]. 

### 4.4. High-Frequency Ultrasound

On the other hand, ultrasonography represents a routine support technique in various branches of medicine, such as gynecology, gastroenterology, cardiology, angiology, and in the head and neck region, to detect structural abnormalities and/or tumors of the salivary glands [132].

This technique for oral cavity exploration is not widely used, but various applications have been reported in the literature: ranging from the study of periodontal and peri-implant tissues to the clinical and surgical characterization of benign and malignant lesions of the oral mucosa [133,134]. Thanks to the advancement of technology, some recent ex vivo and in vivo studies suggest that high-frequency ultrasound holds the ability to distinguish between tissue layers based on image contrast [135]. This contrast represents the result of acoustic backscatter, which is directly proportional to the change in density found at the boundaries between tissue layers. As a result, the layers of less dense adipose tissue appear hypoechoic (black), while denser tissue layers such as muscle and connective tissue appear hyperechoic (white) on images [136]. Intraoral ultrasonography is a diagnostic imaging technique that uses high-frequency sound waves to produce images of structures within the oral cavity [133]. This type of ultrasonography has shown feasibility to be adopted in oral medicine to evaluate the size and position of various structures such as teeth, jaw bones, and soft tissues, as well as to identify pathological conditions such as neoformation, cysts, and abscesses. The procedure constitutes a non-invasive method and does not involve radiation exposure, making it a safe and effective tool for routine dental evaluations [53].

As shown in a study by Iida et al., ultrasonography represents a reproducible and valid technique for the evaluation of a tumor’s Depth of Invasion (DOI), confirming the accuracy of a preoperative measurement of DOI in 56 patients affected by tongue squamous cell carcinoma (TSCC) [137]. Di Stasio et al. presented striking images of squamous cell carcinoma of the tongue, which ultrasonographically presents as a rather defined hypoechoic lesion in relation to the surrounding tissues with homogeneous or relatively hyperechoic echogenicity, emphasizing the great potential for diagnostic support in this type of neoplastic lesions [134].

### 4.5. Photodiagnosis and Photodynamic Therapy

An additional method for early diagnosis and treatment of malignant and potentially malignant lesions of the oral cavity involves the use of photosensitizing substances, irradiated by light through laser and nonlaser devices; the most commonly used device is the diode laser because it is small and inexpensive [138,139]. Siddiqui et al. recently proposed a portable and easy-to-use fiber-coupled LED system that uses a smartphone fluorescence imaging device [140]. Regarding photosensitizing agents, the use of Photofrin^®^, administered intravenously, has been suggested for the treatment of oral cancer at any stage [141]. On the other hand, 5-aminolevulinic acid (5-ALA), administered both topically and systemically, is unable to penetrate tissue so it is more suitable for use in the treatment of premalignant lesions [142].

**Table 1 bioengineering-11-00228-t001:** Summary data of the included studies.

Cit. Num.	First Author	Year of Publication	Title	Methodology	Type of Neoplastic Lesion	Analysed Outcome
[87]	Lucchese A	2005	Proteomic Scan for Tyrosinase Peptide Antigenic Pattern in Vitiligo and Melanoma: Role of Sequence Similarity and HLA-DR1 Affinity1	biomarkers	vitiligo, melanoma	the tyrosinase autoepitope profile of melanoma/vitiligo patient sera
[88]	Tiwari R	2004	Computational peptide dissection of Melan-a/MART-1 oncoprotein antigenicity	biomarkers	melanoma	human Melan-A/MART-1 sequence
[89]	Willers J	2005	Definition of Anti-Tyrosinase MAb T311 Linear Determinant by Proteome-Based Similarity Analysis	biomarkers	vitiligo, melanoma	the human tyrosinase protein sequence
[91]	Gupta A	2018	Role of E-Cadherin in Progression of Oral Squamous Cell Carcinoma: A Retrospective Immunohistochemical Study	biomarkers	OED, OSCC	E-cadherin expression in oral carcinogenesis
[66]	Lo Muzio L	2005	P-Cadherin Expression and Survival Rate in Oral Squamous Cell Carcinoma: An Immunohistochemical Study	biomarkers	OSCC	prevalence of P-cadherin expression in oral squamous cell carcinoma
[92]	Carinci F	2002	CD44 as Prognostic Factor in Oral and Oropharyngeal Squamous Cell Carcinoma	biomarkers	OSCC	CD44 expression
[69]	Pannone G	2007	CYCLOOXYGENASE ISOZYMES IN ORAL SQUAMOUS CELL CARCINOMA: A REAL-TIME RT-PCR STUDY WITH CLINIC P A THOLOGICAL CORRELA TIONS	biomarkers	OSCC	COX-l and COX-2 m-RNA expression
[71]	Aquino G	2012	pEGFR-Tyr 845 expression as prognostic factors in oral squamous cell carcinoma	biomarkers	OSCC	EGFR expression
[97]	Lucchese A	2016	The Potential Role of in Vivo Reflectance Confocal Microscopy for Evaluating Oral Cavity Lesions: A Systematic Review.	in-vivo imaging	OED, OSCC	reflectance confocal microscopy (RCM) for evaluating oral cavity lesions
[100]	Maitland KC	2008	In Vivo Imaging of Oral Neoplasia Using a Miniaturized Fiber Optic Confocal Reflectance Microscope	in-vivo imaging	OSCC	reflectance confocal appearance for evaluating oral cavity lesions
[101]	Agozzino M	2014	Noninvasive, in Vivo Assessment of Oral Squamous Cell Carcinoma	in-vivo imaging	OSCC	reflectance confocal appearance for evaluating oral cavity lesions
[103]	Gentile E	2017	The Potential Role of in Vivo Optical Coherence Tomography for Evaluating Oral Soft Tissue: A Systematic Review	in-vivo imaging	OED, OSCC	optical coherence tomography (OCT) for evaluating oral cavity lesions
[104]	Wilder-Smith P	2009	In Vivo Diagnosis of Oral Dysplasia and Malignancy Using Optical Coherence Tomography: Preliminary Studies in 50 Patients	in-vivo imaging	OED, OSCC	optical coherence tomography (OCT) for evaluating oral cavity lesions
[105]	Heidari AE	2019	Optical Coherence Tomography as an Oral Cancer Screening Adjunct in a Low Resource Settings	in-vivo imaging	OED, OSCC	optical coherence tomography (OCT) for evaluating oral cavity lesions
[106]	Sunny SP	2019	Intra-Operative Point-of-Procedure Delineation of Oral Cancer Margins Using Optical Coherence Tomography	in-vivo imaging	OED, OSCC	optical coherence tomography (OCT) for evaluating oral cavity malignant lesions
[107]	Sengupta S	2018	Review on the Use of Magnetic Fields and Ultrasound for Non-Invasive Cancer Treatment	in-vivo imaging	cancers	effects of magnetic fields and ultrasound on cancer cells and their application for cancer treatment
[108]	Romano A	2022	An Overview of High-Definition Ultrasounds Applied in Oral Diseases	in-vivo imaging	OSCC	High-Definition Ultrasound for evaluating oral cavity lesions
[109]	Di Stasio D	2022	High-Definition Ultrasound Characterization of Squamous Carcinoma of the Tongue: A Descriptive Observational Study	in-vivo imaging	squamous carcinoma of the tongue	High-Definition Ultrasound for evaluating oral cavity lesions
[112]	Iida, Y	2018	Depth of Invasion in Superficial Oral Tongue Carcinoma Quantified Using Intraoral Ultrasonography	in-vivo imaging	oral tongue carcinoma	High-Definition Ultrasound for evaluating Depth of invasion (DOI) in oral carcinoma
[140]	Siddiqui SA	2022	Clinical evaluation of a mobile, low-cost system for fluorescence guided photodynamic therapy of early oral cancer in India.	in-vivo imaging	OED, OSCC	Photodiagnosis (PD) and Photodynamic therapy (PDT) for detection and treatment of early oral cancer

OED = oral epithelial dysplasia, OSCC = oral squamous cell carcinoma.

Protoporphyrin IX (the actual photosensitizer) is produced precisely from 5-ALA in cells capable of heme synthesis and there is a negative feedback mechanism involving ALA synthase, thus preventing overproduction. Unlike other sensitizing substances that induce photosensitivity for several weeks, this agent peaks in 4–6 h and is cleared in about 24 h, without significant collateral effects, making it possible to repeat treatment even at short intervals [143]. On the other hand, the latest generation of photosensitizing agents is conjugated with antibodies, protein/receptor complexes or liposomes, thus presenting a greater affinity for the target tissue [144]. After dosing, the extent of the lesion is first assessed using a blue light that activates red fluorescence (photodiagnosis, PD), with higher intensity in tumor cells due to their ability to produce more protoporphyrin than the surrounding cells.

Photodynamic therapy exploits the same agent illuminated by red light; this light activation triggers a photochemical reaction, leading to the production of reactive oxygen species, thus inducing tissue necrosis [145]. In 1993, as reported by Grant et al., 5-ALA is given orally for systemic use (previously used topically or by intravenous administration) for the first time in the treatment of advanced oral OSCC, inducing necrosis in three out of four patients [143]. This treatment is precise, selective, and noninvasive and can be used for potentially malignant, precancerous, and cancerous lesions even when other therapeutic strategies, such as surgery, are not applicable [140].

Today, the indications for photodynamic therapy have diversified, being effective against viral, bacterial or fungal infections, malignant and pre-malignant lesions of other regions of the body (prostate, bladder, stomach, breast, etc.). It is also used in the treatment of benign and malignant skin diseases, such as Actinic Keratosis, often with the use of 5-ALA, ensuring a good aesthetic outcome among the other benefits [146].

## 5. Limitations of the Study

An analysis of oral pathology focusing on three topics—malignant and premalignant lesions, and diagnostic aids—was conducted. However, due to the heterogeneity of the fields, a systematic search was not feasible; instead, an overview was performed. The substantial amount of data collected precluded meta-analysis, as it comprised a combination of in vivo and ex vivo studies, technological support for in vivo diagnostic imaging, and analyses involving microbiome or salivary assessments.

## 6. Conclusions

Oral Squamous Cell Carcinoma (OSCC) is a type of cancer that affects the cells lining the mouth and can be potentially life-threatening if not detected and treated early. Factors that increase the risk of oral cancer include tobacco use, excessive alcohol intake, UV light exposure, inadequate oral hygiene, and a compromised immune system. Regular dental exams and routine screenings are crucial for early diagnosis. Possible treatments include surgery, radiation, chemotherapy, and targeted therapy. Advances in technology and medical research offer hope for improving OSCC and the accuracy of its precursor entities’ diagnosis and early detection rates in the future. Diagnostic processes have always been crucial in the clinical course; an excellent diagnostic path should guarantee the lowest level of invasiveness, ease and quickness of execution, and the reproducibility of the procedure [25,53]. The potential role of innovative technological and molecular methods does not represent an alternative to histopathological examination [119]. Nonetheless, the current literature agrees with the suitability of these methods for oral tissue inspection, before or during a biopsy and in monitoring the evolution of oral lesions through follow-up appointments. Future investigations are highly desirable to alleviate the current delays in diagnosis and expedite the diagnostic process, ensuring early and non-invasive identification of Oral Malignant and Potentially Malignant Disorders.

## Figures and Tables

**Figure 1 bioengineering-11-00228-f001:**
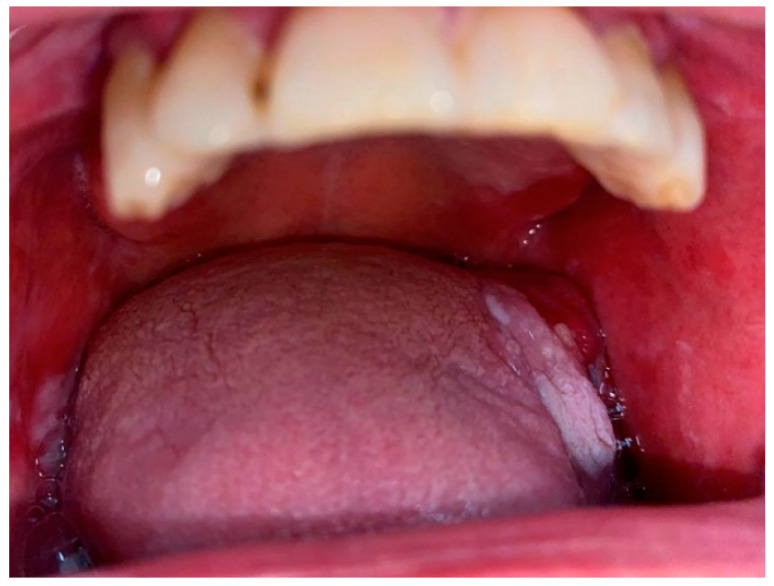
Oral Leukoplakia involving the tongue.

**Figure 2 bioengineering-11-00228-f002:**
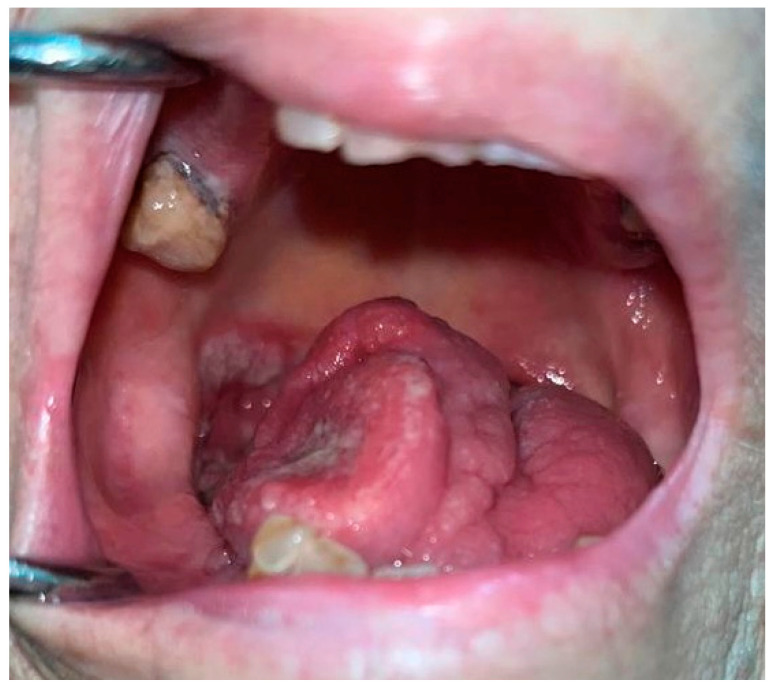
Oral Squamous Cell Carcinoma involving the tongue.

## Data Availability

The original contributions presented in the study are included in the article, further inquiries can be directed to the corresponding author.

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
