# Peer review of "State of the Art in the Diagnosis and Assessment of Oral Malignant and Potentially Malignant Disorders: Present Insights and Future Outlook—An Overview"

_bioengineering, 2024, doi:10.3390/bioengineering11030228_

Round 1

Reviewer 1 Report

Comments and Suggestions for Authors

This is a well written and wide ranging review of the assessment and diagnosis of Oral Potentially Malignant Disorders (OPMD).  However, very little is said about treatment. I would suggest revising the title to reflect this, such as:

State-of-the-art in the diagnosis and assessment of Oral Malignant and Potentially Malignant Disorders: Present Insights and Future Outlook

A wide range of diagnostic options are discussed beyond just examining the oral cavity under direct vision. These include high frequency ultrasound, optical coherence tomography (OCT) and reflectance confocal microscopy (RCM), all of which are undoubtedly useful. Ultrasound is standard and straight forward for many organs, although is relatively new for the oral cavity. OCT and RCM require sophisticated and expensive equipment and a high level of skill to interpret the results. The more molecular tests are also well described.

However, there is a glaring omission. Why is there no mention of photodiagnosis (PD) and photodynamic therapy (PDT)? PD is a simple diagnostic test.  The patient is given a photosensitising drug. This is taken up with some degree of selectivity between malignant and normal tissue. When the area is exposed to blue light, red fluorescence is excited which is brighter in tumour areas than in normal tissue. For superficial lesions, PD can be followed by therapeutic red light (PDT) to ablate the lesion. The efficacy can be followed by photobleaching (lack of fluorescence after red light delivery) and subsequent healing is better than after many other therapeutic options (due to minimal scarring or functional impairment). As with the other techniques described, confirmatory biopsy is required to confirm the diagnosis.

There are many publications on PD and PDT for the detection and treatment of oral cancer since the technique was first reported in 1993 (Photodynamic therapy of oral cancer: photosensitisation with systemic ALA. Grant et al  Lancet.  342:147-148.  1993). A recent publication has shown the potential of PDT for low cost detection and treatment of early oral cancer in rural villages in India. (Clinical evaluation of a mobile, low-cost system for fluorescence guided photodynamic therapy of early oral cancer in India. Siddiqui et al https://doi.org/10.1016/j.pdpdt.2022.102843 ). PDT also has a role in the treatment of more advanced oral cancers. It can sometimes be used when all other treatments (surgery, radiotherapy, chemotherapy) have failed. Minimally invasive, interstitial, image guided PDT can sometimes treat lesions inaccessible to other options.

As the authors say, Treatment options include surgery, radiation therapy, chemotherapy, and targeted therapy. The type and extent of treatment will depend on the size and location of the tumor and the overall health of the patient. It is important to work with a healthcare team to develop an individualized treatment plan. Regular dental check-ups and oral cavity screenings can help detect OSCC in its early stages and increase the chances of successful treatment. The early diagnosis of the asymptomatic first stages of oral cancer is still the pivotal and most demanding step to conveying an adequate clinical outcome in most patients.

I agree with all this, but this review is incomplete without including photodiagnosis (PD) and photodynamic therapy (PDT)

It is stated that: Early detection and treatment, such as cryotherapy or topical medications, can prevent the progression of AK (actinic keratosis) to malignant transformation. The authors should note that PDT is a licensed and widely used treatment for AK in the USA using the photosensitising agent 5-aminolaevulinic acid.

Comments on the Quality of English Language

Only a few minor edits required.

Author Response

Gentle Reviewer,

Thank you for the attention you reserved for our manuscript. 

Comments:

1- As suggested, the title has been revised to be more indicative of what is treated in our work

2- As requested, a paragraph has been added to include photodiagnosis (PD) and photodynamic therapy (PDT) and make our review more comprehensive

Thank you in advance for your kind cooperation and we look forward to hearing 
from you soon.

Kind regards,  

Dr. David GUIDA

Reviewer 2 Report

Comments and Suggestions for Authors

Dear Authors,

Oral malignant and potentially malignant disorders are the important clinical problem. Your

review offers useful knowledge but in this form I am not able to accept it for publication. Please perform following changes:

Mayor changes:

1. Add to the tittle information what kind of review is your study.

2. What was the strategy of study selection? What kind of databases you performed searches? What was exclusion criteria of study?

3. Add study limitations.

Minor changes:

1. “Oral Potentially Malignant Disorders” divide this paragraph into subsection: Leukoplkia, Erythroplakia etc.

2. “Future Perspectives” should be divided into subsection such: biomarkers, reflectance confocal microscopy, reflectance confocal appearance, optical coherence tomography, high-Definition ultrasound etc.

3. “The aim of this study was to examine the available knowledge in literature on the potential use of technology and biomarkers for accurately identifying and categorizing malignant or potentially malignant oral lesions at an early stage.” It should be moved to the end of introduction section.

Best regards,

Reviewer.

Author Response

Gentle Reviewer,

Thank you for the attention you reserved for our manuscript. 

Comments:

1- As suggested, the title has been revised to be more indicative of what is treated in our work.

2,3- We initiated our study with an analysis of trending topics in oral pathology, focusing on malignant and premalignant lesions, as well as diagnostic aids. However, due to the inherent heterogeneity within these fields, conducting a systematic search proved unfeasible. Consequently, we opted for an overview approach. The primary limitations of our study stem from the considerable heterogeneity observed in the obtained results, which precluded meta-analytical analysis. Our data comprise a diverse mix of in vivo and ex vivo studies, along with technological support for in vivo diagnostic imaging, as well as analyses involving microbiome and salivary assessments. Despite our efforts, conducting meta-analytic statistics was not feasible.

Minor changes have all been made.

Thank you in advance for your kind cooperation and we look forward to hearing 
from you soon.

Kind regards,  

Dr. David GUIDA

Reviewer 3 Report

Comments and Suggestions for Authors

Please rewrite for readability.

Comments on the Quality of English Language

Some grammar is inappropriate please rewrite for readability.

Author Response

Gentle Reviewer,

Thank you for the attention you reserved for our manuscript. 

Comments:

As suggested, we have revised our manuscript for better readability.

Thank you in advance for your kind cooperation and we look forward to hearing 

from you soon.

Kind regards,  

Dr. David GUIDA

Round 2

Reviewer 1 Report

Comments and Suggestions for Authors

The authors have responded well to my main main criticism, which was the omission of any reference to PD & PDT. It is clear that the authors are not familiar with these techniques and the comments on them are relatively superficial, but the basic facts are OK. No PDT publications are included in the comprehensive list of other publications in table 1 and there are a couple of typing errors on the PDT refs (Grant paper was 1993, not 1983). Other PDT paper was Siddiqui, not Shahid).

With these minor edits, I think this paper is now acceptable for publication

Comments on the Quality of English Language

Occasional minor edits only

Author Response

Gentle Reviewer,

Thank you for the attention you reserved for our manuscript. 

Comments:

1- As suggested, Siddiqui paper has been added to the table 1.

2- As highlighted, the two typing errors have been corrected. I apologize for them.

I want to really thank you for your suggestions, they were invaluable and I hope they can contribute to publication of our work.

Best regards,

Dr. David GUIDA

Reviewer 2 Report

Comments and Suggestions for Authors

Dear Authors,

All of suggested corrections have been applied.

Best regards,

Reviewer.

Dear Authors,

All of suggested correctins have been applied.

Best regards,

Author Response

Gentle Reviewer,

Thank you for the attention you reserved for our manuscript. Your suggestions were precious and I hope they can contribute to its publication.

Best regards,

Dr. David GUIDA

Reviewer 3 Report

Comments and Suggestions for Authors

Thank you for the revisions.

Author Response

(The authors gave the same response as above.)
